# Two-Step Targeted Drug Delivery via Proteinaceous Barnase-Barstar Interface and Doxorubicin-Loaded Nano-PLGA Outperforms One-Step Strategy for Targeted Delivery to HER2-Overexpressing Cells

**DOI:** 10.3390/pharmaceutics15010052

**Published:** 2022-12-24

**Authors:** Elena N. Komedchikova, Olga A. Kolesnikova, Ekaterina D. Tereshina, Polina A. Kotelnikova, Anna S. Sogomonyan, Alexey V. Stepanov, Sergey M. Deyev, Maxim P. Nikitin, Victoria O. Shipunova

**Affiliations:** 1Moscow Institute of Physics and Technology, 141701 Dolgoprudny, Russia; 2Shemyakin-Ovchinnikov Institute of Bioorganic Chemistry, Russian Academy of Sciences, 117997 Moscow, Russia; 3Nanobiomedicine Division, Sirius University of Science and Technology, 354340 Sochi, Russia

**Keywords:** HER2, targeted chemotherapy, scaffold proteins, DARPin, PLGA, polymer particles

## Abstract

Nanoparticle-based chemotherapy is considered to be an effective approach to cancer diagnostics and therapy in modern biomedicine. However, efficient tumor targeting remains a great challenge due to the lack of specificity, selectivity, and high dosage of chemotherapeutic drugs required. A two-step targeted drug delivery strategy (DDS), involving cancer cell pre-targeting, first with a first nontoxic module and subsequent targeting with a second complementary toxic module, is a solution for decreasing doses for administration and lowering systemic toxicity. To prove two-step DDS efficiency, we performed a direct comparison of one-step and two-step DDS based on chemotherapy loaded PLGA nanoparticles and barnase*barstar interface. Namely, we developed and thoroughly characterized the two-step targeting strategy of HER2-overexpressing cancer cells. The first targeting block consists of anti-HER2 scaffold polypeptide DARPin9_29 fused with barstar. Barstar exhibits an extremely effective binding to ribonuclease barnase with K_aff_ = 10^14^ M^−1^, thus making the barnase*barstar protein pair one of the strongest known protein*protein complexes. A therapeutic PLGA-based nanocarrier coupled to barnase was used as a second targeting block. The PLGA nanoparticles were loaded with diagnostic dye, Nile Blue, and a chemotherapeutic drug, doxorubicin. We showed that the two-step DDS increases the performance of chemotherapy-loaded nanocarriers: IC50 of doxorubicin delivered via two-step DDS was more than 100 times lower than that for one-step DDS: IC50 = 43 ± 3 nM for two-step DDS vs. IC50 = 4972 ± 1965 nM for one-step DDS. The obtained results demonstrate the significant efficiency of two-step DDS over the classical one-step one. We believe that the obtained data will significantly change the direction of research in developing targeted anti-cancer drugs and promote the creation of new generation cancer treatment strategies.

## 1. Introduction

Cancer is one of the most significant threats to humankind. In 2020, 19.1 million cases were registered, and this number is supposed to reach 28.4 million by 2040. It is the second most frequent cause of death, and the contribution of cancer to the mortality rate continues to grow [1]. Conventional cancer therapy strategies suffer from a lack of selectivity and low drug efficiency, and are frequently associated with side effects including cardiac dysfunction, cytopenia, infection, diarrhea, vomiting, and others [2,3].

Chemotherapy-loaded nanoparticles capable of increasing the amount of therapeutic drug that reaches the tumor site and reducing the systemic toxicity provide encouraging solutions to the described problems. Moreover, drug encapsulation into nanoparticle architecture can increase the bioavailability of the chemotherapeutic compound, extend the duration of action via bloodstream circulation prolongation, and can solve problems associated with hydrophobicity and insolubility of drugs [4,5,6,7,8].

Several nanoparticle-based medications, such as PEGylated liposomal doxorubicin Caelyx [9], and liposomal formulation of daunorubicin and cytarabine VYXEOS [10], have already been approved by FDA for cancer treatment. However, despite the improved efficiency of such formulations, the delivery of nanoparticles occurs by passive transport through enlarged pores in the vascular endothelium of tumors known as the “enhanced permeability and retention (EPR)” effect [4]. However, it was shown that the EPR effect fails in some tumors and in some patients. Therefore, it is critical to develop different approaches for the delivery of nanoparticles to cancer cells [11]. One of the ways to implement targeted delivery is to modify the surface of nanoparticles with molecules binding to certain cancer cell receptors. This is a rapidly developing branch of biomedicine that has already demonstrated several promising results in clinical trials [12,13]. Various proteins such as antibodies, transferrin, EGF, lectins, non-immunoglobulin scaffold proteins, as well as protein-nucleic acid complexes, aptamers, and small molecules like folic acid and sugars, are traditionally used for targeted drug delivery [14,15,16,17,18]. Currently, small synthetic polypeptides (scaffold molecules) emerge as the most promising targeting compounds due to their remarkable affinity, stability, ease of biotechnological production, and the absence of immunomodulation in vivo [14,19,20,21,22,23,24,25].

A pre-targeting concept implying two-step delivery of therapeutic compounds to the tumor site is expected to provide significant systemic nanoparticle toxicity reduction and nanoparticle targeting abilities improvement [26,27,28]. This concept is based on the initial delivery of the first targeting non-toxic compound to specific cancer cells in a moderately high dose (thus realizing pre-targeting) followed by the delivery of a relatively small dose of a second toxic compound interacting with the first one in a key and lock mode. A two-step drug delivery strategy (DDS) offers a series of benefits over standard one-step strategies, such as (1) reduced toxicity for normal cells; (2) controlled penetration of toxin into the tumor; (3) improved drug biodistribution; (4) reduction of the required dose of the drug [29]. The disadvantages of the currently available two-step DDSs are due to the immunogenicity of the components, possible competition with molecules in the bloodstream, and the expensive and time-consuming biotechnological production in mammals of the components of such DDSs [30,31].

Here, we describe the application of the barnase*barstar pair for the two-step delivery of doxorubicin-loaded PLGA nanoparticles to HER2-overexpressing cancer cells. Namely, we showed that the barnase*barstar-mediated two-step drug delivery strategy (two-step DDS) significantly outperforms the one-step delivery strategy (one-step DDS) using doxorubicin-loaded polymer nanoparticles directed toward HER2 receptor on cancer cells.

Ribonuclease barnase and its natural inhibitor barstar are small proteins (12 and 10 kDa) of bacterial origin that are not presented in mammals and possess an extremely high constant of binding (K_aff_ = 10^14^ M^−1^) [22,32,33,34,35]. We synthesized polymer PLGA nanoparticles and loaded them with the chemotherapeutic drug doxorubicin and the fluorescent dye Nile Blue, and successfully modified the surface of nanoparticles with barnase. The barnase*barstar interface was used as “Lego” bricks to link the toxic PLGA nanoparticles with scaffold protein DARPin9_29 recognizing the tumor marker HER2 on the surface of cancer cells. DARPin9_29 was genetically fused with barstar to obtain Bs-DARPin9_29 protein capable of specifically targeting HER2-overexpressing cancer cells. We showed two-step efficient labeling of HER2-overexpressing cancer cells with supramolecular structure PLGA-Barnase*DARPin9_29-Barstar self-assembled on the cell surface.

The developed two-step HER2-overexpressing cell targeting strategy was studied in detail and thoroughly characterized with different physicochemical methods. The direct comparison of one-step DDS, two-step DDS, and delivery of free doxorubicin in molecular form was carefully performed thus proving the superior efficiency of the two-step DDS over one-step DDS.

## 2. Materials and Methods

### 2.1. Nanoparticle Synthesis

PLGA nanoparticles were synthesized by the double emulsion “water-oil-water” method, followed by solvent evaporation, according to a modified procedure developed by us earlier [36,37,38]. The first emulsion was obtained by adding 150 μL of an aqueous solution of doxorubicin hydrochloride at a concentration of 2 g/L to 300 μL of PLGA at a concentration of 40 g/L in chloroform and 50 μL of Nile Blue Blue at various concentrations in chloroform, followed by sonication for 1 min at 40% amplitude and for 1 min at 60% amplitude using a 130 W ultrasonic disintegrator (Sonics, New Town, CT, USA) at +4 °C. The second emulsion was created by mixing the first emulsion with 3 mL of 5% polyvinyl alcohol (PVA) solution in milliQ with the addition of 1 g/L chitosan oligosaccharide lactate. The solution was sonicated for 1 min at 40% amplitude and 1 min at 60% amplitude at +4 °C. The resulting solution was incubated with slow shaking for chloroform evaporation, then washed three times in PBS (137 mM NaCl, 2.7 mM KCl, 4.77 mM Na_2_HPO_4_·2H_2_O, 1.7 mM KH_2_PO_4_, pH 7.4) by centrifugation, and resuspended in 300 μL of PBS. The final concentration of nanoparticles was determined by drying at +60 °C and followed by weighing the dry residue.

### 2.2. Electron Microscopy

Electron microscopy images of PLGA nanoparticles loaded with Nile Blue and doxorubicin were obtained with a MAIA3 (Tescan, Brno, Czech Republic) microscope at an accelerating voltage of 15 kV. The samples were deposited onto a silicon wafer and then dried in the air. The resulting images were processed using ImageJ software to obtain a particle size distribution.

### 2.3. Particle Size and Surface Charge Measurements

The hydrodynamic sizes and ζ-potential of PLGA nanoparticles loaded with Nile Blue and doxorubicin were determined in PBS at 25 °C using a Zetasizer Nano ZS (Malvern Instruments Ltd., Malvern, UK) analyzer.

### 2.4. Nanoparticle Tracking Analysis

The size of nanoparticles and their distribution were measured with nanoparticle tracking analysis. It was performed with an AstraTrace (Abisense, Sochi, Russia) equipped with an sCMOS scientific camera (PRIME 95B, Teledyne Photometrics, Tucson, AZ, USA) and 405 nm violet laser with regulated power (24 mW power was set in the experiments). Nanoparticles were analyzed in PBS. The videos were recorded at 6.6 fps with 100 ms exposure time. 1000 frames were collected for each sample. The data were processed with AstraTrace software using trackpy [39] algorithm. The minimal length of trajectory was set to 5. Approximately 100 particles per frame we observed during the recording.

### 2.5. Nanoparticles Conjugation with Proteins

Barnase and barstar were expressed and purified as described by us previously [32]. PLGA nanoparticles were covalently modified with barnase or barstar proteins using EDC (1-ethyl-3-(3-dimethylaminopropyl)carbodiimide hydrochloride) and sulfo-NHS (*N*-hydroxysulfosuccinimide sodium salt) as cross-linking agents through the formation of amide bonds between amino groups from chitosan on the particle surface and protein carboxyl groups. 200 μg of protein was activated by a 15-fold molar excess of EDC and sulfo-NHS in 0.1 M MES (2-(*N*-morpholino)ethanesulfonic acid), pH 5.0 for 40 min at room temperature. Then protein was added to 1 mg of PLGA nanoparticles in (1) borate buffer—0.4 M H_3_BO_3_, 70 mM Na_2_B_4_O_7_·10H_2_O, pH 8.0, (2) 0.1 M HEPES (4-(2-hydroxyethyl)-1-piperazineethanesulfonic acid) pH 6.0, and sonicated. The mixture was incubated for at least 5 h at room temperature, periodically treated in an ultrasonic bath, and the unbound protein was washed off by triple centrifugation for 5 min at 8000× *g* in PBS with 1% BSA (bovine serum albumin).

### 2.6. Measurement of Barnase Activity

The RNAse activity of barnase was investigated by the method of the acid-insoluble precipitate [40]. The protein solution or suspension of nanoparticles in 40 μL of buffer (0.125 M Tris-HCl, pH 8.5) was mixed with 160 μL of yeast RNA at a concentration of 2 g/L and incubated at 37 °C for 15 min. The reaction was terminated by the addition of 200 μL of 0.625 N H_2_SO_4_, and the mixture was incubated for 5 min at room temperature. Undigested RNA was separated by centrifugation at 14,000× *g* for 15 min at room temperature. Optical density was measured at λ = 260 nm (OD260) corresponding to the concentration of free mononucleotides and proportional to the activity of the enzyme (ribonuclease). The inhibition of RNAse activity of barnase was measured similarly: nanoparticles conjugated with barstar were pre-incubated with 2.5 nM barnase, and the enzymatic activity of the mixture was measured as described above.

### 2.7. Fluorescent Spectroscopy

Excitation and emission spectra of PLGA nanoparticles loaded with Nile Blue and doxorubicin were obtained using an Infinite M100 Pro (Tecan) microplate reader. Nanoparticle suspension at a concentration of 10 μg/mL in 100 μL of PBS was placed in a 96-well flat-bottomed plate. Excitation spectra were measured in a range from 350 to 675 nm (with the emission of 700 nm), and emission spectra from 675 to 850 nm (excitation wavelength 650 nm). The stability of PLGA nanoparticles’ fluorescence at a final concentration of 1 g/L was measured several times during one week at an excitation of 570 nm and emission of 690 nm both in H_2_O and DMSO (dimethyl sulfoxide).

### 2.8. Protein Conjugation to Nanoparticles

Bn-DARPin9_29 was expressed and purified as described by us previously [41]. DARPin 9_29 fused with barstar was cloned into pET22b using barstar from the plasmid pIG-4D5-Barnase-Barstar [32] with the following protein sequence: (DARPin9_29)-([Gly_4_Ser]_3_)-(Barstar)-(His_5_-tag), thus making N-term of DARPin9_29 and C-term of barstar (through His_5_-tag) available for chemical conjugation or other manipulations. Freshly transformed Escherichia coli BL1 (DE3) strain was grown in ZYM-5052 autoinduction medium (2% tryptone, 1% yeast extract, 0.5% glycerol, 0.05% glucose, 0.02% lactose, 25 mM Na_2_HPO_4_, 25 mM KH_2_PO_4_, 50 mM NH_4_Cl, 5 mM Na_2_SO_4_, 2 mM MgSO_4_ [42]) containing 0.2 g/L ampicillin. The culture was grown in a thermostatic shaker at 25 °C and 200 rpm overnight. Cells were harvested by centrifugation at 5000× *g* for 15 min at 4 °C. Cell pellets were resuspended in lysis buffer (20 mM Na-Pi, 300 mM NaCl, pH 7.4, 50 μg/mL lysozyme) and then sonicated on ice. Cellular debris was removed by centrifugation at 30,000× *g* at +4 °C for 2 h. After the addition of imidazole (20 mM), the supernatant was filtered through a 0.22 μm membrane and applied onto a HisTrap HP, 1 mL column (GE Healthcare, Chicago, Illinois, USA) equilibrated with 20 mM sodium phosphate buffer (pH 7.4), 300 mM NaCl and 20 mM imidazole. The bound proteins were eluted with an imidazole step gradient (50, 75, 100, 150, 200, 250, and 500 mM). The fractions were analyzed by 15% reducing SDS-PAGE.

Proteins were conjugated to FITC as follows: 100 μg of protein in 90 μL of PBS was rapidly mixed with 10 μL of FITC in DMSO and incubated overnight at room temperature with an 8-fold molar excess of FITC to protein. Proteins were purified from unreacted FITC molecules using Zeba Spin Desalting Columns, 7k MWCO (Pierce, Waltham, MA USA) according to the manufacturer’s recommendations.

### 2.9. Cell Culture

Cell lines: human breast adenocarcinoma cells SK-BR-3, Chinese hamster ovary cells CHO, were cultured in DMEM medium supplemented with 10% fetal bovine serum, penicillin-streptomycin, and 2 mM L-glutamine. Cells were incubated in a humidified atmosphere with 5% CO_2_ at 37 °C. Cells were passaged when they reached 80–90% of the monolayer. To remove cells from the surface of the plastic, 2 mM EDTA (ethylenediaminetetraacetic acid) solution in PBS was used without trypsin addition (to avoid disruption of the integrity of cell receptors). Cell lines were maintained in culture for no more than two months, after which they were replaced with fresh frozen lines. For cell counting, a Countess (Invitrogen, Waltham, MA, USA) automatic cell counter was used. For this, 5 μL of 0.4% trypan blue, which stains only dead cells, was added to 5 μL of cell suspension. The solution was pipetted and added to the slides for automatic cell counting.

### 2.10. Flow Cytometry

To determine the efficiency of cell labeling with FITC-modified proteins, the cell suspension was washed with PBS and resuspended in 300 μL of PBS with 1% BSA at a concentration of 10^6^ cells/mL. Cells were labeled with proteins in a final concentration of 2 μg/mL, washed, and analyzed in the FL1 channel (excitation laser—488 nm, emission filter—530/30 nm) using a BD Fortessa (BD, Holdrege, NE, USA) flow cytometer for Trastuzumab-FITC and DARPin-based proteins.

To determine the specificity of targeted nanoparticles, the cell suspension was washed in PBS and resuspended in 300 μL of PBS with 1% BSA at a concentration of 10^6^ cells/mL. Cells were incubated with Bs-DARPin9_29 at concentrations 0, 0.5 and 2.5 μg/mL for 1 h at +4 °C, washed from unbound protein by triple centrifugation. Then, PLGA-Bn conjugates were added to the cells, incubated for 15 min, and washed from unbound nanoparticles by double centrifugation. Samples were analyzed using a BD Accuri C6 (BD) flow cytometer in the FL4 detection channel (excitation laser—644 nm, emission filter—675/25 nm).

### 2.11. Cytotoxicity Assay

The cytotoxicity of the synthesized nanoparticles and doxorubicin was investigated using a standard MTT-test. Cells were seeded in a 96-well plate at 10^4^ cells per well in 100 μL of full DMEM with an addition of 10% FBS (fetal bovine serum) and cultured overnight. Then, test substances were added to the wells in 100 μL of DMEM and incubated for 72 h. Next, the medium was removed, and 100 μL of MTT (tetrazolium dye 3-(4,5-dimethylthiazol-2-yl)-2,5-diphenyltetrazolium bromide) solution at a concentration of 0.5 g/L in DMEM was added to the cells. The samples were incubated for 1 h at 37 °C, the MTT solution was removed, and 100 μL of DMSO was added to the wells. The plate was gently shaken until the formazan crystals were completely dissolved. The optical density of each well was measured with an Infinite M1000 Pro microplate analyzer (Tecan) at a wavelength of 570 nm and a reference wavelength of 630 nm. The viability of cells was presented in percent in comparison to the untreated sample. All samples were performed in triplicate. IC_50_ was calculated using GraphPad Prism software v8.0.1 (244).

## 3. Results

### 3.1. Design of the Experiment: Comparison of One-Step DDS with Two-Step DDS

The main goal of this work was to demonstrate the higher efficiency of the two-step targeted drug delivery strategy (DDS) in comparison with one-step DDS; we implemented the following experimental scheme. Polymer PLGA nanoparticles loaded with the fluorescent dye, Nile Blue, and the chemotherapeutic drug, doxorubicin, were thoroughly characterized with different physicochemical methods. These nanoparticles carrying the chemotherapeutic drug were used for the targeted delivery to HER2-overexpressing cells using two different strategies (Figure 1):(1)Conventional targeting strategy, hereinafter referred as one-step DDS was performed as follows. PLGA-based polymer nanoparticles were synthesized and chemically functionalized with an anti-HER2 scaffold protein, namely DARPin9_29.(2)The delivery of chemotherapeutic drug through the pre-targeting scheme, hereinafter referred as two-step DDS was mediated via the barnase*barstar proteinaceous interface.

Both strategies, one-step DDS and two-step DDS, were directly compared in terms of cytotoxicity to HER2-overexpressing cells. We examined the targeted delivery efficiency and calculated the half-maximal inhibitory concentration of doxorubicin loaded into PLGA nanoparticles (IC50) delivered via one-step DDS or two-step DDS. The direct comparison of these two strategies showed that two-step DDS for doxorubicin is significantly more effective than one-step DDS (Figure 1).

### 3.2. Synthesis of Polymer Nanoparticles Based on Poly-(D,L-Lactic-Co-Glycolic Acid)

Poly-(D,L-lactic-co-glycolic acid) nanoparticles (PLGA) for imaging and therapeutic applications were synthesized as shown in Figure 2. Nile Blue ([9-(diethylamino)benzo[a]phenoxazin-5-ylidene] azanium sulfate, also known as Nile Blue A) was incorporated into the nanoparticles as a fluorescent dye that allows to track the particles inside cells and use them for diagnostic purposes. This dye is successfully used in a wide range of biological applications, such as gel electrophoresis, staining of histological sections, labeling of neutral lipids and fatty acids, and visualization of cancer cells [43]. Nile Blue is a biocompatible dye with absorption and emission maxima in the near-infrared optical window (635 and 674 nm in water solution, respectively), which makes it optimal for labeling target cells in vitro and in vivo. Doxorubicin was incorporated into nanoparticles that allow using them for therapeutic applications. Doxorubicin is an anthracycline antibiotic that causes cell death by the interaction with DNA and inhibition of topoisomerase II, which leads to the suppression of nucleic acids synthesis and the formation of free radicals that destroy cellular membranes and biomolecules [44,45].

Nanoparticles were synthesized by the double emulsion “water-oil-water” method with subsequent evaporation of the solvent, as shown in Figure 2. The first emulsion was obtained by the addition of water doxorubicin solution to the solution of PLGA and Nile Blue in chloroform, followed by a short sonication. The second emulsion was obtained by the addition of the first emulsion to the PVA solution containing 1 g/L of chitosan oligosaccharide lactate, followed by the second short sonication. After chloroform evaporation by the slow mixing, nanoparticles were centrifugated and resuspended in PBS.

Since chitosan oligosaccharide lactate is insoluble in chloroform, and PLGA and chitosan are dissolved in different phases during the synthesis (PLGA in chloroform, chitosan oligosaccharide lactate is in water), therefore, these polymers do not mix with each other. Since PLGA polymer is negatively charged, and chitosan oligosaccharide lactate is positively charged, this synthesis method makes it possible to obtain nanoparticles consisting mainly of PLGA polymer and electrostatically coated with chitosan for further modifications. Moreover, it was shown that chitosan coating facilitates the interaction of nanoparticle with cells and prevent the rapid uptake of nanoparticles in the mononuclear phagocyte system [46,47]. The as-synthesized chitosan-coated PLGA nanoparticles are hereinafter referred to as PLGA nanoparticles.

### 3.3. Characterization of PLGA Nanoparticles

The morphology of as-synthesized nanoparticles loaded with Nile Blue and doxorubicin was studied by scanning electron microscopy (Figure 3a). The data presented in Figure 3a illustrate that synthesized PLGA nanoparticles are spherical monodisperse structures. Image processing shows that the average size and standard deviation of PLGA nanoparticles loaded with doxorubicin and Nile Blue are 218 ± 59 nm (Figure 3b).

The hydrodynamic size of nanoparticles, measured by the dynamic light scattering method, was found to be 201 ± 38 nm (Figure 3c and Figure 4a), which is surprisingly similar to the value of the physical size of nanoparticles determined by SEM (218 ± 59 nm). The ζ-potential of nanoparticles, measured by the electrophoretic light scattering method, was –1.6 ± 0.8 mV (Figure 3c and Figure 4b) thus slightly deviating from zero. Such surface charge at pH 7.4 was due to the presence of both negatively charged carboxyl groups –COOH (within the composition of PLGA) and positively charged amino groups –NH_2_ (within the composition of chitosan) on the surface of the nanoparticles. Measurements of the hydrodynamic size and ζ-potential of nanoparticles loaded with Nile blue and doxorubicin after 1.5 years of storage at +4 °C in PBS showed that the particles did not form aggregates, slightly increased in size, and ζ-potential remained virtually unchanged thus proving colloidal stability of nanoparticles (Figure 3c).

The effective incorporation of the fluorescent dye Nile Blue was investigated by fluorescence spectroscopy by measuring the excitation and fluorescence emission spectra of nanoparticles. The excitation spectra were measured in the range from 350 to 675 nm (with emission measured at 700 nm). Four PLGA nanoparticles with different Nile Blue concentrations used in the synthesis were investigated. The excitation and emission spectra (Figure 5) demonstrate that the most effective Nile Blue concentration during the synthesis is 1.7 g/L, the further scaling up of Nile Blue concentration leads to a decrease in fluorescence intensity. It is most probably caused by non-fluorescent H-aggregate formation with an absorption shifted to the blue region of the spectrum. The nanoparticle tracking analysis showed the size of nanoparticles to be equal to 231 nm, 227 nm, 189 nm, 181 nm for particles with 5 g/L, 1.7 g/L, 0.5 g/L, 0.18 g/L of Nile Blue used in the synthesis, respectively (Appendix A). These data show that the Nile Blue concentration significantly affects the nanoparticle size and must be considered when developing scalable methods for the synthesis of nanoparticles for in vivo injections.

The efficient incorporation of doxorubicin was investigated by fluorescent spectroscopy on nanoparticles that do not contain Nile Blue. Nanoparticles were dissolved in DMSO and then fluorescence was measured using a fluorescence calibration curve for doxorubicin samples in the same solutions (Figure 6a). The measurement of the fluorescence of the samples showed that doxorubicin incorporation was 0.9 nmol doxorubicin per 1 mg of nanoparticles. The stability of fluorescence of the synthesized nanoparticles loaded with doxorubicin and Nile Blue was investigated for 1 week, both in water and DMSO; the data presented in Figure 6b confirm that the particles do not bleach during storage at room temperature in a plastic tube without light protection at least for one week (no further observations were performed).

### 3.4. Barnase*Barstar Protein Interface for the Two-Step DDS: Chemical Modification of PLGA Nanoparticles with Adaptor Proteins

One of the central problems of modern chemotherapy is its relative non-specificity. The surface of nanoparticles is modified with targeting molecules to deliver them to specific cells and tissues. This can help to reduce the non-specific toxicity of drugs to normal non-transformed cells. To make this kind of modification universal for any target on the cell surface and include the possibility to “cancel the action on demand”, we propose to mediate the interaction between toxic nanoparticles and molecules recognizing cancer cells using protein adaptors, the barnase*barstar protein pair. Barstar (10 kDa) is a natural inhibitor of bacterial ribonuclease barnase (12 kDa). The N- and C-terms of both proteins are available for chemical conjugation and genetic engineering and are not located in the active site of both enzymes [32,41].

We used scaffold protein DARPin9_29, which recognizes the receptor HER2 on the surface of cancer cells with high affinity (K_D_ = 3.8 nM), for the targeted delivery of synthesized polymer PLGA nanoparticles to cancer cells. This modular DDS based on PLGA nanoparticles, protein adaptors barnase*barstar, and scaffold proteins is schematically illustrated in Figure 1.

The surface of the nanoparticles was modified by one of the components of the pair—Figure 1 shows PLGA nanoparticles covalently modified with barnase. During the pre-targeting process, Bs-DARPin9_29 bifunctional protein was added to the cells with HER2 overexpression leading to the selective binding of the anti-HER2 molecule to the cancer cell surface. Next, self-assembly with the second component of the pair was carried out, namely, with PLGA nanoparticles conjugated with barnase. The resulting supramolecular structure selectively interacted with the cells with overexpression of receptor HER2: DARPin9_29 mediated the internalization of PLGA nanoparticles in the cells, while chemotherapy drug-induced cell death.

Chemical modification of PLGA nanoparticles was carried out using the sodium salt of 1-ethyl-3- (3-dimethyl aminopropyl) carbodiimide, EDC, and the sodium salt of N-hydroxysulfosuccinimide, sulfo-NHS, as cross-linking agents through the formation of amide bonds between the carboxyl groups of proteins and amino groups on the surface of nanoparticles. The amino groups are presented on the surface of nanoparticles because of chitosan oligosaccharide lactate surface stabilization. In the first stage of the reaction, proteins were activated with EDC/sulfo-NHS mixture in an acidic buffer with pH 5.0, then nanoparticles were added to the buffer with pH 6.0 or pH 8.0.

The efficacy of the conjugation of PLGA nanoparticles to barnase was measured by the enzymatic ability of conjugated nanoparticles, namely, their ability to hydrolyze RNA due to the presence of functionally active barnase on the nanoparticle surface. The measurement was performed by the commonly used method of the acid-insoluble precipitate [40]. First, the solution of conjugated PLGA nanoparticles was mixed with yeast RNA and incubated at 37 °C to digest RNA. Then, the reaction was stopped by the addition of sulfuric acid, and the supernatant containing uncleaved RNA was separated by centrifugation. The optical density of the solution corresponding to the concentration of free mononucleotides and proportional to the activity of the enzyme was measured by the microplate reader in 96-well UV-Vis transparent plates. The value of sample absorbance at 260 nm is proportional to the concentration of free mononucleotides in the solution, thus reflecting the RNAse activity of the tested sample, either ribonuclease activity of the tested proteins or nanoparticles conjugated to proteins.

The efficiency of the conjugation of PLGA nanoparticles to barstar was measured in a similar way by testing the ability of barstar to inhibit the RNAse activity of free barnase added to the sample of nanoparticles. Nanoparticles conjugated with barstar were pre-incubated with barnase, and the enzymatic activity of the mixture was measured as described above.

First, we tested the functional activity of free barnase and barstar before conjugation to nanoparticles. The enzymatic activity of free barnase and barstar proteins is shown in Figure 7a. The purple curve corresponds to the activity of the free barnase and has a concentration-dependent manner achieving saturation. As a positive control in the experiment directed towards the investigation of the enzymatic activity of PLGA nanoparticles conjugated with barnase, a sample of free barnase at a concentration of 2.5 nM was used. This point corresponds to the middle of the linear range of the barnase enzymatic activity curve. The enzymatic activity of barstar, namely the ability to inhibit barnase, was investigated similarly by its pre-incubation with 2.5 nM of barnase (green curve in Figure 7a) and measuring the enzymatic activity of the sample. As a positive control in the investigation of the enzymatic activity of PLGA nanoparticles conjugated with barstar, a sample with barstar at a concentration of 15 nM (+ barnase 2.5 nM) was used.

We obtained three types of PLGA nanoparticles modified with barnase by three different methods: (i) carbodiimide conjugation at pH 8.0, (ii) carbodiimide conjugation at pH 6.0, (iii) non-covalent protein adsorption on the particle surface. For the modification via protein adsorption, PLGA nanoparticles were incubated with barnase or barstar in PBS at the same concentrations used for covalent coupling for 5 h at room temperature.

Data presented in Figure 7b indicate that the highest efficiency of modification of PLGA nanoparticles with barnase is achieved during chemical conjugation at pH 6.0. Similar data were obtained for PLGA nanoparticles conjugated with barstar: the highest inhibition of the barnase activity is achieved for conjugates obtained at pH 6.0. Therefore, the possibility of obtaining functionally effective PLGA nanoparticles in terms of enzymatic activity with both barnase and barstar has been demonstrated.

### 3.5. Two-Step Targeted Delivery of Polymer PLGA Nanoparticles to the HER2-Overexpressing Cells through the Barnase*Barstar Interface

We demonstrated the efficiency of two-step DDS for the targeted delivery to cancer cells. For cancer cell pre-targeting, DARPin9_29 fused with barstar was selected since this protein, in contrast to DARPin9_29 fused with barnase (Appendix A), does not exhibit significant cytotoxicity and can be further used in vivo as a pre-targeting molecule in large doses without any significant side effects. In contrast to DARPin9_29 fused with barstar, DARPin9_29 fused with barnase exhibit its own anti-cancer effect [48] and is not the best option for the pre-targeting strategy implying the use of first non-toxic component injected in a large dose before the injection of the second toxic component.

Namely, conjugates of PLGA nanoparticles with barnase, PLGA-Bn, were obtained and self-assembled on the cancer cell surface using barstar fused with DARPin9_29 [49,50,51]. Thus, supramolecular structures PLGA-Bn*Bs-DARPin9_29 were assembled on the cell surface using the pre-targeting concept via Bs-DARPin9_29 protein and subsequent binding with PLGA-Bn. These structures were delivered to the cells overexpressing receptor HER2.

For the cell culture experiments, we selected two cell lines with various levels of HER2 expression, namely SK-BR-3 and CHO cells. SK-BR-3 is a mammary adenocarcinoma cell line with overexpression of HER2 (about 10^6^ receptors per cell), while CHO, Chinese hamster ovary cells, do not express any receptor of the EGFR family. Expression of the HER2 receptor on these cells was confirmed by confocal microscopy (Figure 8) and by flow cytometry (Figure 9) by imaging cells with fluorescently labeled full-length antibodies against HER2—Trastuzumab-FITC. Also, the binding of DARPin9_29 was confirmed by cell labeling with DARPin9_29-FITC (Figure 9). Data from confocal microscopy and cytometry assays presented in Figure 8 and Figure 9 indicate that SK-BR-3 cells do express HER2 and effectively labeled with full-length anti-HER2 antibody Trastuzumab and anti-HER2 scaffold protein DARPin9_29.

The functional activity of DARPin within the composition of fusion proteins with barnase and barstar, Bn-DARPin9_29 and Bs-DARPin9_29, was confirmed by flow cytometry (Figure 9) by labeling cells with HER2 receptor overexpression. It was demonstrated that the presence of barnase and barstar in fusion protein does not affect the interaction of DARPin9_29 with HER2-positive cells. Thus, both components of the barnase*barstar protein pair did not influence the functional activity of recognizing scaffold DARPin9_29 and can be used as adaptor proteins mediating two-step targeted drug delivery.

The functional polymer PLGA nanostructures were used for the selective targeting of cells with HER2 overexpression. Labeling was carried out by the two-step targeted delivery method. In the first step, cells were incubated in suspension with Bs-DARPin9_29 in two different concentrations, followed by washing from unbound protein. Next, the cells were labeled by PLGA-Bn, followed by washing from unbound particles. The binding between nanostructures and cells was estimated by flow cytometry with excitation with a 640 nm laser in the fluorescence channel corresponding to the Nile Blue fluorescence.

The data presented in Figure 10a indicates highly effective labeling of HER2-overexpressing cells by polymer PLGA nanostructures, assembled on the cells’ surface, PLGA-Bn*Bs-DARPin9_29. Non-specific labeling of cells by PLGA-Bn conjugates is not observed, and binding of PLGA-Bn*Bs-DARPin9_29 with cells has a concentration-dependent manner. With an increase in the concentration of Bs-DARPin9_29 by 5 times from 0.5 μg/mL to 2.5 μg/mL, the median fluorescence intensity of the cell population increases by 17,551/2904 = 6 times.

### 3.6. Cytotoxicity Study: Comparison of One-Step DDS and Two-Step DDS

The synthesized PLGA polymer nanoparticles contain a fluorescent dye, Nile Blue, and a chemotherapeutic drug, doxorubicin, which induces cell death via apoptosis.

To directly compare the efficiency of one-step DDS and two-step DDS, the standard cytotoxicity assay was performed. The standard MTT-test was performed three days after the addition of the nanostructures in different concentrations to the HER2-overexpressing cells.

The therapeutic efficacy of two-step DDS was compared with: (i) free doxorubicin, which was added to the cells, (ii) non-targeted PLGA loaded with doxorubicin; (iii) one-step DDS, namely, PLGA directly conjugated to DARPin9_29 (Figure 10b).

Half-maximal inhibitory concentration (IC50) of doxorubicin calculated for different formulations was found to be:(i)Free doxorubicin: IC50 = 441 ± 61 nM;(ii)Non-targeted PLGA: IC50 = 134 ± 51.2 nM;(iii)One-step DDS: IC50 = 4972 ± 1965 nM;(iv)Two-step DDS: IC50 = 43 ± 3 nM.

The quantitative analysis of obtained results demonstrates that two-step DDS is much more effective than one-step DDS in terms of IC50. Namely, cells exposed to non-targeted PLGA nanoparticles or one-step DDS were not affected by the cytotoxic properties of doxorubicin loaded inside nanoparticles, and cells survived by more than 82% even at the highest concentrations of PLGA, namely, at 1 g/L (Figure 10b). However, the delivery of doxorubicin via two-step DDS decreased its IC50 by 10.3 times vs. free doxorubicin and more than 100 times vs. one-step DDS.

Usually, the incorporation of doxorubicin into nanoparticle structure does not significantly decrease its IC50 or even increase it [52,53]. The main goal of such incorporation is to minimize the cardiac toxicity of doxorubicin and increase its tumor accumulation [54,55,56]. However, for targeted nanoparticles this trend is not always observed. It was previously shown that the decoration of PLGA nanoparticle surface with targeting peptides can decrease the IC50 of doxorubicin by more than 1 order of magnitude. Namely, the IC50 was shown to be equal to 5745 ± 2651 nM, 92 ± 55 nM, and 331 ± 37 nM for non-targeted doxorubicin-loaded PLGA, anti-EGFR PLGA, and free doxorubicin, respectively [57]. Another example showed that anti-HER2 chitosan nanoparticles loaded with doxorubicin demonstrate IC50 equal to 377 nM in contrast to 1235 nM for free doxorubicin [58]. Such unexpected results may be explained by the alterations in mechanisms of doxorubicin action in cells when it is delivered with a nanocarrier.

The primary doxorubicin toxicity mechanism is based on the intercalation with DNA base pairs in the nucleus, thus blocking topoisomerase II and causing DNA breakage, leading to the inhibition of DNA, RNA, and protein biosynthesis processes [45]. However, when doxorubicin is delivered within the targeted nanoparticle to cell-membrane receptors, other mechanisms of its cytotoxicity may become more pronounced. Namely, the oxidation of doxorubicin to semiquinone with the release of reactive oxygen species leading to lipid peroxidation and membrane damage is most likely to have a more pronounced cytotoxic effect when it occurs near the cell membrane with the slow release of doxorubicin from nanoparticles in contrast to doxorubicin freely dispersed in the cytoplasm and nucleus as a result of diffusion processes.

Hence, based on the results obtained, incorporation of a chemotherapeutic drug in the composition of polymer PLGA nanoparticles assembled on the surface of the cancer cells via barnase*barstar interface significantly decreases the concentration of chemotherapeutic drug doxorubicin, needed to receive the same cytotoxic effect in comparison with either doxorubicin in molecular form or one-step DDS.

## 4. Discussion

Here, we describe the development and characterization of the two-step DDS based on polymer PLGA nanoparticles possessing both diagnostic and therapeutic properties. The delivery of these nanoparticles to HER2-overexpressing cancer cells was realized via the proteinaceous barnase*barstar interface and HER2-recognizing scaffold protein DARPin9_29. We have shown that this two-step DDS significantly outperforms the traditional one-step DDS for HER2-overexpressing cell elimination.

The development of targeted nanoparticle-based DDS is one of the most rapidly developing directions of modern biomedicine. Nanoparticles exhibit unique properties as therapeutic compounds and often outperform their molecular counterparts. The use of full-size IgG for targeted delivery to cells is already becoming outdated in favor of artificial scaffold polypeptides. The small size, high stability, and ease of biotechnological production allow scaffolds to be effectively used as tools for targeted delivery [16,24]. In particular, the use of scaffolds (for example, DARPins, 14–16 kDa) makes it possible to equip the surface of nanoparticle by targeting molecules with high density in contrast to, e.g., big IgG (150 kDa) to get an optimal affinity in the nanoparticle*cell interaction. On the other hand, the small size of DARPins conjugated to the surface of therapeutic nanoparticles can lead to steric hindrance during target recognition.

In particular, we previously performed a direct comparison of HER2-overexpressing cell-targeting abilities of nanoparticle*DARPin9_29 and nanoparticle*anti-HER2 IgG for two nanoparticle types: SiO_2_ fluorescent nanoparticles and SiO_2_ magnetic nanoparticles [41]. We showed that direct conjugation of DARPin9_29 to nanoparticle surface is absolutely ineffective for HER2 targeting in contrast to nanoparticle*anti-HER2 IgG. However, cell targeting mediated with proteinaceous barnase*barstar interface showed a significantly higher efficiency compared to both nanoparticle*DARPin9_29 and nanoparticle*anti-HER2 IgG. The nanoparticle cell uptake was more than 3 times higher for the two-step approach in comparison with the one-step one [41]. Thus, the barnase*barstar interface significantly enhanced the particle binding efficiency and increased the cell uptake under equal conditions.

These in vitro results led us to the concept of cancer cell pre-targeting: the enhancement of a targeted nanomedicine specific toxicity can be achieved through the pre-targeting of the tumor site with a large amount of a targeted non-toxic compound that increases the affinity of the sequentially injected second nanoparticle-based cytotoxic module. This strategy makes it possible, using the same administered doses of nanoparticles, to achieve a significantly higher efficiency of nanoparticles both for diagnostics and therapy. Different pre-targeting strategies are now under investigation, namely those based on streptavidin*biotin pair [59,60,61], antibody-hapten interactions [62,63], DNA, RNA, or mirror-imaged oligonucleotides [64,65,66,67,68], click-chemistry based platforms or SpyTag/SpyCatcher systems [69]. Some of these strategies possess serious disadvantages, such as sterical hindrance during the target recognition, the significant difference in the size of both components, and high immunogenicity or irreversibility of binding. However, the barnase*barstar protein pair outperforms others two-step DDS which makes it a unique tool for the design of multifunctional biomedical products. Barstar (10 kDa) is a natural inhibitor of bacterial ribonuclease barnase (12 kDa) [32]. These proteins have an extremely high binding affinity (association constant K_aff_ ~ 10^14^ M^−1^) and fast interaction kinetics (rate constant of complex formation k_on_ ~ 10^8^ M^−1^s^−1^).

Previously we showed the versatility of the barnase*barstar protein pair for a wide range of applications, including targeted delivery of protein molecules, nanoparticles, and different supramolecular structures. In particular: (i) the successful labeling of HER2-overexpressing cancer cells in vitro with the self-assembled structures consisting of the magnetic particles and quantum dots using barstar and scFv-barnase-scFv construct (directed toward HER2 antigen) [31] and in vivo with radiolabeled 4D5 scFv–barnase and 4D5 scFv–dibarnase [32] were shown; (ii) a universal delivery system based on barnase*barstar and SiO_2_-binding peptide was developed [41]; (iii) bispecific antibodies against HER1 and HER2 antigens using 425scFv-barstar and 4D5scFv-barnase [70] were obtained and utilized for imaging of cancer cells with overexpression of these receptors [71].

Considering the potential in vivo applications of barnase*barstar protein pair for diagnostic and therapeutic applications, it should be emphasized that this system significantly outperforms other protein complementary systems in different aspects. Namely, it was previously shown that this protein system possesses unique stability under severe conditions (low pH, high temperature, and presence of chaotropic agents) in contrast to other known complementary protein systems (streptavidin*biotin, antibody*antigen, and protein A*immunoglobulin) [34]. This unique stability makes us believe that this complex will not be degraded in the bloodstream, where much milder conditions are realized, and will allow the self-assembly of different structures in vivo. More importantly, these proteins are not presented in mammals, thus making them excellent candidates for use in the bloodstream without any interaction with endogenic components of blood in contrast to, e.g., streptavidin*biotin pair; biotin is a B7 vitamin that is presented in blood in a relatively high amount [33,72]. Moreover, these proteins possess unique proteolytic stability and are not subject to rapid degradation by proteases in the bloodstream. In contrast, various systems based on DNA/RNA self-assembly undergo rapid degradation due to nucleases presented in serum.

These features made possible the successful use of the barnase*barstar protein pair for the in vivo delivery of radiolabeled anti-HER2 trimer complexes, namely 4D5 scFv–barnase*barstar, in Balb/c Nu/Nu mice reaching more than 8% of anti-HER2 complex accumulation in HER2-positive tumors [32]. Moreover, it was shown that the barnase*barstar pair is an effective tool for switchable targeting of solid HER2-positive tumors by CAR-T cells in vivo [48], thus reasonably confirming the efficacy and versatility of the barnase*barstar interface for the wide range of biological applications in vitro and in vivo that require the self-assembly of different structures in different conditions.

Here, using Barstar-DARPin9_29 and Barnase-conjugated polymer PLGA nanoparticles loaded with fluorescent dye Nile Blue and chemotherapeutic drug doxorubicin, we showed successful labeling and killing of HER2-overexpressing cells. The use of such two-step DDS allows decreasing the necessary dose of the doxorubicin needed to cause cancer cell death: the incorporation of doxorubicin in the composition of targeted two-step DDS nanoparticles decreases its IC50 by 10.3 times vs. free doxorubicin and more than 100 times vs. one-step DDS.

## 5. Conclusions

Here, we reported the versatile method of the two-stage drug delivery strategy (DDS) for theranostic applications based on the barnase*barstar proteinaceous interface. The small size and high affinity constant of these proteins make them an excellent “molecular glue” for the design of different self-assembling structures based on various modules, where one component of this DDS is in the structure of one module (e.g., Barnase in the therapeutic module), and another strategy component in the structure of another module (e.g., Barstar in targeted DARPin module). This “lego” approach allows escaping such chemical conjugation issues, such as non-oriented molecule modification of the nanoparticle surface, denaturation of proteins on the nanoparticle surface, conjugation through several functional groups on the same molecule, and the impossibility of simple replacement of nanoparticle composition. On the contrary, using the proposed platform for nanoparticle biomodification allows obtaining biologically active structures by either mixing components, such as nanoparticle-Barnase + Barstar-DARPin, or two-step targeted delivery in vitro and in vivo. Such a pre-targeting concept can significantly reduce the doses of drugs (incorporated into the second toxic component of DDS) to obtain the same therapeutic effect, thus reducing side effects and systemic toxicity. We believe that the proposed strategy outperforms the existing technologies and will promote the development of new-generation drug delivery strategies for cancer diagnostics and treatment.

## Figures and Tables

**Figure 1 pharmaceutics-15-00052-f001:**
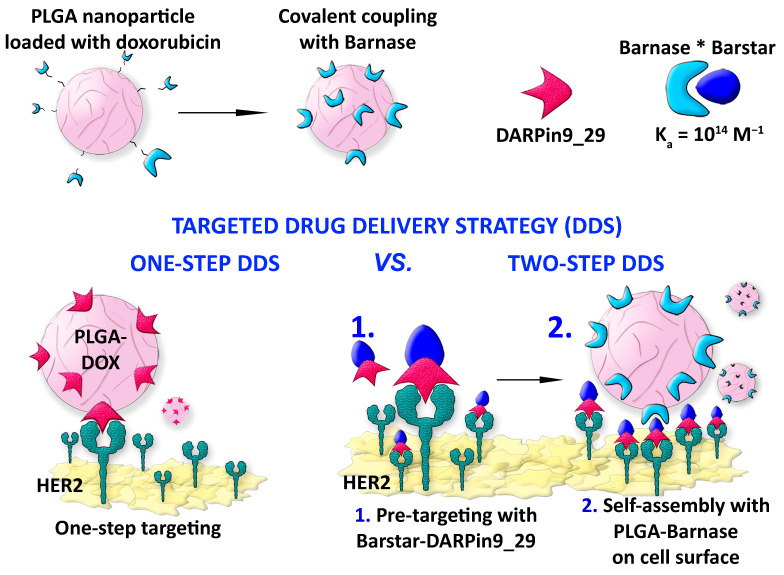
Scheme of the experiment. Protein-assisted one-step DDS and two-step DDS. Polymer nanoparticles loaded with the chemotherapeutic drug doxorubicin were delivered to HER2-overexpressing cancer cells. For the one-step DDS, PLGA particles were covalently modified with targeting scaffold protein DARPin9_29 that selectively recognizes receptor HER2 on the surface of cancer cells. For two-step DDS, the PLGA surface was covalently modified with barnase; then, the structure was assembled with barstar fused with DARPin9_29. The self-assembly of nanoparticles with scaffold polypeptides using protein interface barnase*barstar for the delivery to HER2-overexpressing cancer cells significantly increases the therapeutic efficacy of chemotherapy-loaded nanoparticles.

**Figure 2 pharmaceutics-15-00052-f002:**
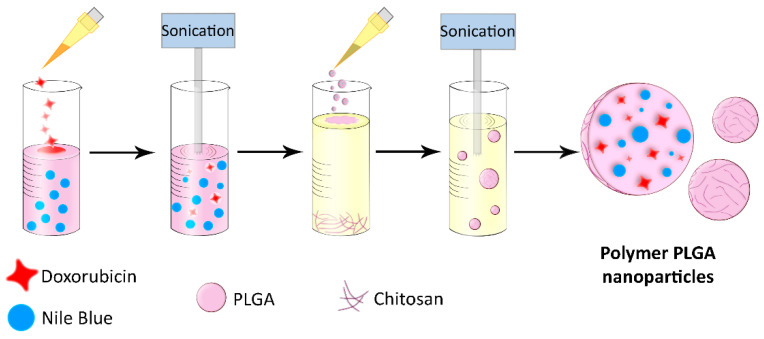
Schematic illustration of polymer nanoparticle synthesis via double emulsion method. The first emulsion was produced by sonication of doxorubicin water solution and solution of PLGA and Nile Blue in chloroform. The second emulsion was created by adding the first emulsion to the PVA and chitosan oligosaccharide lactate water solution. Next, the as-obtained suspension was centrifuged thrice and thus colloidally stable polymer PLGA nanoparticles were obtained.

**Figure 3 pharmaceutics-15-00052-f003:**
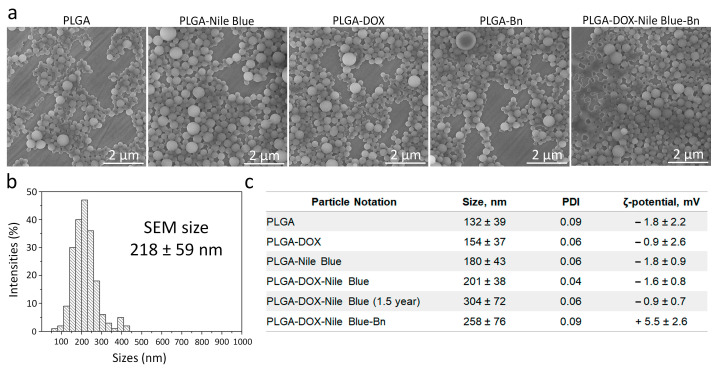
Characterization of PLGA nanoparticles. (**a**) Scanning electron microscopy analysis of pristine polymer nanoparticles (PLGA), particles loaded with Nile Blue only (PLGA-Nile Blue), particles loaded with doxorubicin only (PLGA-Dox), particles conjugated with barnase (PLGA-Bn), particles loaded with doxorubicin, Nile Blue and conjugated with barnase (PLGA-DOX-Nile Blue-Bn). (**b**) The physical size distribution of nanoparticles loaded with Nile Blue and doxorubicin is obtained by image processing. (**c**) Hydrodynamic sizes, polydispersity indices, and ζ-potentials of pristine PLGA nanoparticles, particles loaded with doxorubicin only (PLGA-Dox), particles loaded with Nile Blue only (PLGA-Nile Blue), particles loaded with doxorubicin and Nile Blue (PLGA-Dox-Nile Blue), particles loaded with doxorubicin and Nile Blue (PLGA-Dox-Nile Blue) and stored for 1.5 years, particles loaded with doxorubicin and Nile Blue and conjugated with barnase (PLGA-Dox-Nile Blue-Bn).

**Figure 4 pharmaceutics-15-00052-f004:**
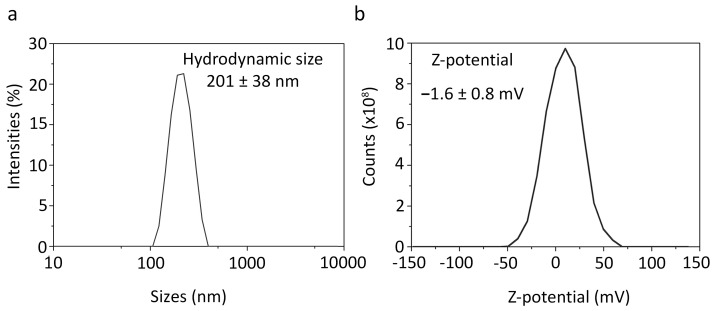
Electrophoretic and hydrodynamic analysis of nanoparticles. (**a**) Hydrodynamic size distribution of nanoparticles loaded with doxorubicin and Nile Blue was obtained by the dynamic light scattering method. (**b**) ζ-potential distribution of nanoparticles obtained by the electrophoretic light scattering method.

**Figure 5 pharmaceutics-15-00052-f005:**
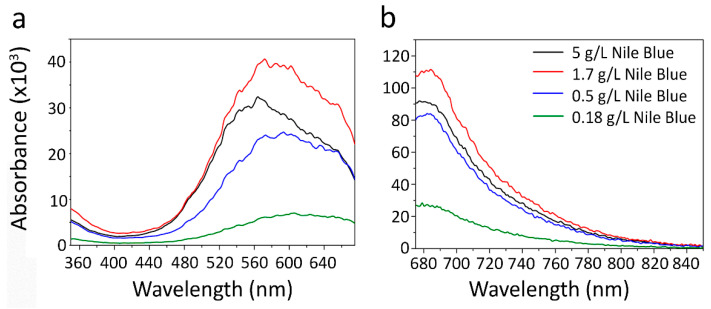
Fluorescence analysis of nanoparticles. (**a**) The excitation spectra (emission wavelength 700 nm) of PLGA nanoparticles according to the fluorescent dye concentration. (**b**) The emission spectra (excitation wavelength 650 nm) of PLGA nanoparticles according to fluorescent dye concentration.

**Figure 6 pharmaceutics-15-00052-f006:**
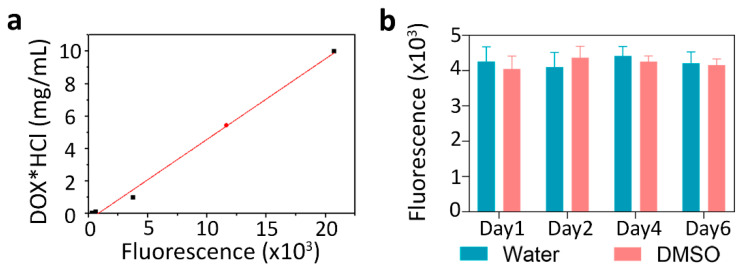
Characterization of PLGA nanoparticles. (**a**) Calibration curve for the measurement of doxorubicin*HCl incorporation into PLGA nanoparticles obtained with fluorescence spectroscopy (excitation 480 nm, emission 590 nm). The red dot indicates the measured doxorubicin concentration in a 1 g/L sample of PLGA particles. (**b**) The stability of fluorescence of PLGA particles loaded with doxorubicin and Nile Blue both in water and in the organic solvent, DMSO for 6 days.

**Figure 7 pharmaceutics-15-00052-f007:**
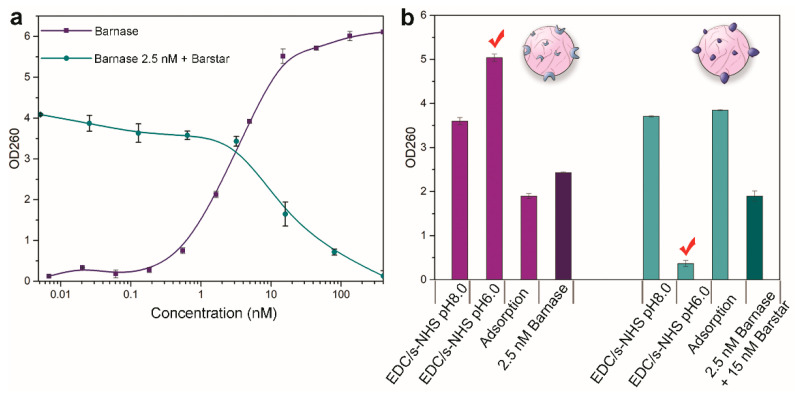
Functional activity of PLGA nanoparticles conjugated with barnase and barstar proteins. (**a**) The enzymatic activity of barnase (purple curve) and barstar (green curve) versus protein concentration. Optical density at 260 nm corresponds to the concentration of free mononucleotides obtained by barnase digestion of yeast RNA (for barnase activity) and by barstar pre-incubated with 2.5 nM of barnase (for barstar inhibition activity). (**b**) The enzymatic activity of PLGA conjugates with barnase (purple bars) and barstar (green bars). Data is presented for conjugates obtained at pH 8.0, pH 6.0, and by non-covalent protein absorption on the particle surface. Optical density (OD260) corresponded to the concentration of free mononucleotides was measured at a wavelength of 260 nm. The “√” symbol refers to the conjugation methods that were selected for further work.

**Figure 8 pharmaceutics-15-00052-f008:**
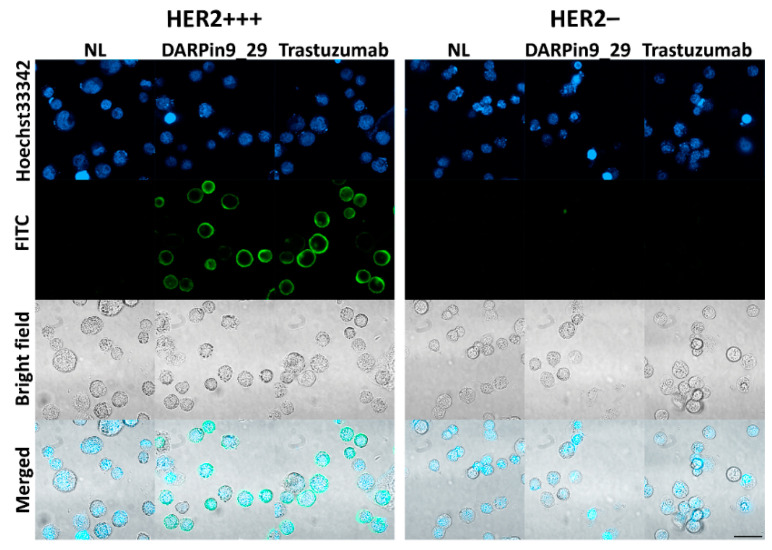
Confocal laser scanning microscopy of SK-BR-3 and CHO cells, labeled with DARPin9_29-FITC and Trastuzumab-FITC. Bottom panels—bright-field images, top panels—fluorescent images of the cells labeled with FITC-conjugated proteins, and Hoeschst33342. Scale bar, 50 µm.

**Figure 9 pharmaceutics-15-00052-f009:**
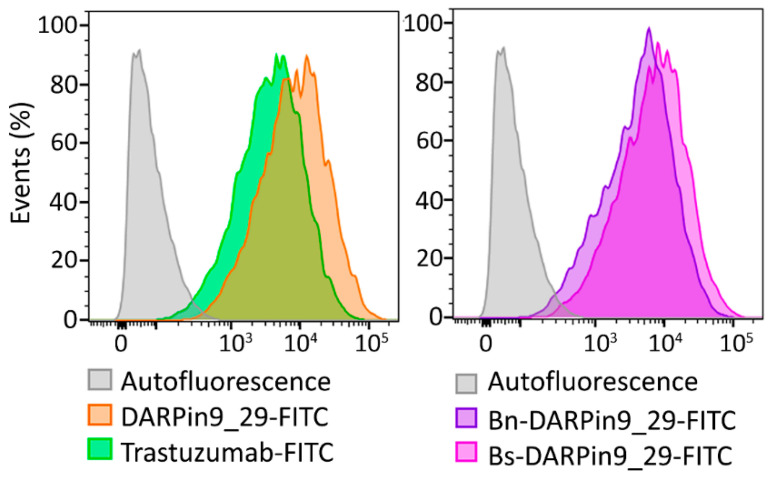
Flow cytometry of SK-BR-3 cells, labeled with DARPin9_29-FITC, Trastuzumab-FITC, Bn-DARPin9_29-FITC, and Bs-DARPin9_29-FITC in the fluorescence channel FL1 corresponding to FITC fluorescence.

**Figure 10 pharmaceutics-15-00052-f010:**
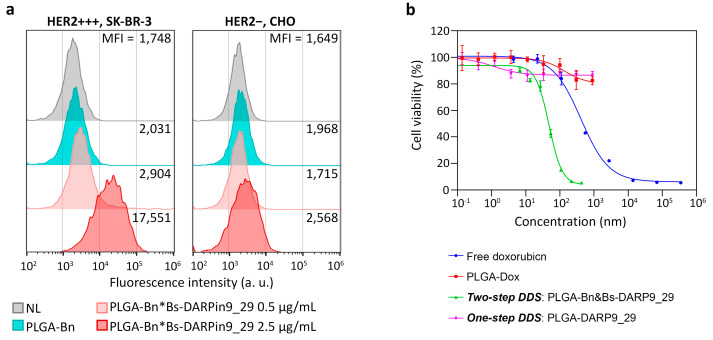
Targeted delivery and cytotoxicity of two-step DDS, PLGA-Bn*Bs-DARPin9_29. (**a**) Flow cytometry of SK-BR-3 and CHO cells, labeled with PLGA-Bn*Bs-DARPin9_29 nanostructures in the fluorescence channel FL4 corresponding to Nile Blue fluorescence. (**b**) Cytotoxicity of free doxorubicin, non-targeted PLGA loaded with doxorubicin, and targeted one-step DDS (PLGA-DARPin9_29) and two-step DDS (PLGA-Bn*Bs-DARPin9_29) based on PLGA particles, obtained with the MTT assay.

## Data Availability

All data are presented within the manuscript and Appendix A.

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
