# Peer review of "Two-Step Targeted Drug Delivery via Proteinaceous Barnase-Barstar Interface and Doxorubicin-Loaded Nano-PLGA Outperforms One-Step Strategy for Targeted Delivery to HER2-Overexpressing Cells"

_pharmaceutics, 2022, doi:10.3390/pharmaceutics15010052_

Round 1

Reviewer 1 Report

The article of Komedchikova et al. entitled "Two-Step Targeted Drug Delivery via Proteinaceous Barnase-Barstar Interface and Doxorubicin-Loaded Nano-PLGA Outperforms One-Step Strategy for Targeted Delivery to HER2-Overexpressing Cells” describes the application of PLGA-Chitosan nanoparticles loaded with doxorubicin and surface-decorated with Proteinaceous Barnase-Barstar proteins for the treatment of cancer. The manuscript is well written, the topic is original and the interest to a general audience. Although some interesting results have been presented, several questions need to be addressed before this manuscript could be considered for publication. I would recommend a resubmission with minor revision based on the following comments:

1.      Line 107. The nanoparticle is formed by a mixture of PLGA and Chitosan (2.5%) polymers. Throughout the text it is only mentioned that nanoparticles are only made of PLGA, and it causes confusion, because it is really made of a mixture. Please could the authors clarify this issue in the text.

2.      Line 138. Please change “ amino groups on the particle surface “ by “amino groups from chitosan on the particle surface”.

3.      Line 285. Please could the authors add the Polydispersity Index (PdI) of the particle size distributions. The value gives us an idea about the uniformity of nanoparticles.

4.      The quality of Figure 3 is very poor. I strongly recommend to the author to perform TEM microscopy with negative staining.

5.      Line 298. The authors state that the nanoparticles are stable in solution because the size values match in DLS and microscopy. However, this is not correct since colloidal stability is predicted by the values of zeta potential and size over time. My question then would be, if the authors have carried out a study of the stability of the nanoparticles over time?

6.      Line 301. Please could the authors add the standard deviation of zeta potential value. Which is the zeta potential value of Nile blue-loaded nanoparticles?

7.      Line 333. Which is the particle size, zeta potential and morphology of Doxorubicin/PLGA, Protein /PLGA and Protein /Doxorubicin/PLGA/nanoparticles? The characterization of the different prototypes of the nanoparticles used in this study is lacking.

8.      Line 374. Line 138. Please change “amino groups on the surface“ by “amino groups from chitosan on surface”.

9.      In Figure S1 and 4 the x-axis should be expressed in Log10 to show all the particle size distribution.

Author Response

Comment 1

Line 107. The nanoparticle is formed by a mixture of PLGA and Chitosan (2.5%) polymers. Throughout the text it is only mentioned that nanoparticles are only made of PLGA, and it causes confusion, because it is really made of a mixture. Please could the authors clarify this issue in the text.

Reply 1

We thank the reviewer for this comment. Indeed, the synthesis process is not described in sufficient detail, which leads to a misunderstanding of the particle composition. Since PLGA and chitosan oligosaccharide lactate generally cannot be dissolved in the same phase (both in chloroform or both in water), this synthesis method allows to get nanoparticles that mainly consist of PLGA and are coated with chitosan oligosaccharide lactate, and the excess of chitosan oligosaccharide lactate in water phase is removed by centrifugation after synthesis. The issue is described in the main text as follows:

“Since chitosan oligosaccharide lactate is insoluble in chloroform, and PLGA and chitosan are dissolved in different phases during the syntheses (PLGA in chloroform, chitosan oligosaccharide lactate is in water), therefore these polymers do not mix with each other. Since PLGA polymer is negatively charged, and chitosan oligosaccharide lactate is positively charged, this synthesis method makes it possible to obtain nanoparticles consisting mainly of PLGA polymer and electrostatically coated with chitosan for further modifications. Moreover, it was shown that chitosan coating facilitates the interaction of nanoparticle with cells and prevent the rapid uptake of nano-particles with mononuclear phagocyte system [46,47]. The as-synthesized chitosan-coated PLGA nanoparticles, hereinafter referred to as PLGA nanoparticles”.

Comment 2

Line 138. Please change “ amino groups on the particle surface “ by “amino groups from chitosan on the particle surface”.

Reply 2

Corrected.

Comment 3

Line 285. Please could the authors add the Polydispersity Index (PdI) of the particle size distributions. The value gives us an idea about the uniformity of nanoparticles.

Reply 3

Corrected. The PdI values for all the studied nanoparticles are added to the table (Fig. 3c). The data presented in Fig. 3c indicate that the polydispersity indices do not exceed 0.1, thereby confirming the uniformity of particles at all stages of synthesis.

Comment 4

The quality of Figure 3 is very poor. I strongly recommend to the author to perform TEM microscopy with negative staining.

Reply 4

Corrected. Indeed, the microelectron images of polymer nanoparticles were not clear enough. Since high voltage transmission electron microscopy results in particles rapidly degrading under the influence of an electron beam, and different contrasting methods often lead to a change in particle morphology and size, we obtained high-resolution microelectron photographs with scanning electron microscopy. Moreover, the morphology of all nanoparticles used in the work was studied, which is summarized in Fig. 3.

Figure 3. Characterization of PLGA nanoparticles. (a) Scanning electron microscopy analysis of pristine polymer nanoparticles (PLGA), particles loaded with Nile Blue only (PLGA-Nile Blue), particles loaded with doxorubicin only (PLGA-Dox), particles conjugated with barnase (PLGA-Bn), particles loaded with doxorubicin, Nile Blue and conjugated with barnase (PLGA-DOX-Nile Blue-Bn). (b) The physical size distribution of nanoparticles loaded with Nile Blue and doxorubicin is obtained by image processing. (c) Hydrodynamic sizes, polydispersity indices and ζ-potentials of pristine PLGA nanoparticles, particles loaded with doxorubicin only (PLGA-Dox), particles loaded with Nile Blue only (PLGA-Nile Blue), particles loaded with doxorubicin and Nile Blue (PLGA-Dox-Nile Blue), particles loaded with doxorubicin and Nile Blue (PLGA-Dox-Nile Blue) and stored for 1.5 year, particles loaded with doxorubicin and Nile Blue and conjugated with barnase (PLGA-Dox-Nile Blue-Bn).

Comment 5

Line 298. The authors state that the nanoparticles are stable in solution because the size values match in DLS and microscopy. However, this is not correct since colloidal stability is predicted by the values of zeta potential and size over time. My question then would be, if the authors have carried out a study of the stability of the nanoparticles over time?

Reply 5

Corrected. Indeed, the claim about the stability of nanoparticles was invalid. Considering that the experiments were carried out 1.5 years ago and the nanoparticle samples were stored at +4 °C in the fridge, we investigated the stability of the nanoparticles loaded with Nile blue and doxorubicin after 1.5-year storage in PBS. The data presented in Fig. 3c indicate that the particles did not aggregate during such a long period of time and remained colloidally stable, and the ζ-potential remained unchanged. The issue is described in the main text as follows:

“The hydrodynamic size of nanoparticles, measured by the dynamic light scattering method, was found to be 201 ± 38 nm (Fig. 4) which is surprisingly quite similar to the value of the physical size of nanoparticles determined by SEM (218 ± 59 nm). The ζ-potential of nanoparticles, measured by the electrophoretic light scattering method, was –1.6 ± 0.8 mV (Fig. 4) thus slightly deviating from zero. Such surface charge at pH 7.4 was due to the presence of both negatively charged carboxyl groups –COOH (within the composition of PLGA) and positively charged amino groups –NH2 (within the composition of chitosan) on the surface of the nanoparticles. Measurements of the hydrodynamic size and ζ-potential of nanoparticles loaded with Nile blue and doxorubicin after 1.5 years of storage at +4 °C in PBS showed that the particles did not form aggregates, slightly increased in size, and ζ-potential remained unchanged thus proving colloidal stability of nanoparticles (Fig. 3c)”.

Comment 6

Line 301. Please could the authors add the standard deviation of zeta potential value. Which is the zeta potential value of Nile blue-loaded nanoparticles?

Reply 6

Corrected. The standard deviation of ζ-potential value was added to the Fig. 4. The ζ-potential of Nile Blue loaded nanoparticles is -1.8 ± 0.9 mV, while ζ-potential of Nile Blue and doxorubicin loaded nanoparticles is -1.6 ± 0.8 mV (Fig. 3c, Fig. 4). All ζ-potentials of particles used in the work are summarized in Fig. 3c.

Comment 7

Line 333. Which is the particle size, zeta potential and morphology of Doxorubicin/PLGA, Protein /PLGA and Protein /Doxorubicin/PLGA/nanoparticles? The characterization of the different prototypes of the nanoparticles used in this study is lacking.

Reply 7

Corrected. The thorough characterization of the different prototypes was performed and summarized in Fig. 3.

Comment 8

Line 374. Line 138. Please change “amino groups on the surface“ by “amino groups from chitosan on surface”.

Reply 8

Corrected.

Comment 9

In Figure S1 and 4 the x-axis should be expressed in Log10 to show all the particle size distribution.

Reply 9

Corrected as follows:

Figure 4. Electrophoretic and hydrodynamic analysis of nanoparticles. (a) Hydrodynamic size distribution of nanoparticles loaded with doxorubicin and Nile Blue was obtained by the dynamic light scattering method. (b) ζ-potential distribution of nanoparticles obtained by the electrophoretic light scattering method.

Figure S1. Physico-chemical properties of PLGA nanoparticles. Particle size distribution obtained with an AstraTrace (Abisense, Russia) for nanoparticles synthesized with 5 g/L (a), 1.7 g/L (b), 0.5 g/L (c), and 0.17 g/L (d) Nile Blue.

Reviewer 2 Report

In this work, the authors developed a two step targeting strategy for doxorubicin delivery. It was interesting to found that two step drug delivery system could increase the performance of drug for more than 100 times. Overall, the topic of this work was interesting, but some specific issues should be addressed before further consideration.

1. the abbreviation of the drug delivery system was suggested to consistent with the one step and two step delivery strategy for better understanding.

2. one of the most important question. The targeting efficiency of wo step targeting delivery strategy was limited by the pretargeting and the second targeting abilities, and the recognition and the targeting efficiency could not reach 100 %. I think one step targeting strategy with anti-HER2 modification could achieve an improved targeting efficiency.

3. Two step targeting delivery strategy faced a limited in vivo application. The authors were suggested to give a detail discussion.

4. The writing should be thoroughly improved, and the manuscript was really hard to understand.

5. The anti-proliferation abilities of the pretargeting moieties should be evaluated.

Author Response

Comment 1

The abbreviation of the drug delivery system was suggested to consistent with the one step and two step delivery strategy for better understanding.

Reply 1

Corrected. The abbreviations are introduced as follows:

“These nanoparticles carrying chemotherapeutic drug were used for the targeted deliv-ery to HER2-overexpressing cells using two different strategies (Fig. 1):

1) conventional targeting strategy, hereinafter referred as one-step DDS was performed as follows. PLGA-based polymer nanoparticles were synthesized and chemically functionalized with an anti-HER2 scaffold protein, namely DARPin9_29.

2) the delivery of chemotherapeutic drug through the pre-targeting scheme, hereinafter referred as two-step DDS was mediated via the barnase*barstar proteinaceous interface”

and used throughout the manuscript.

Comment 2

One of the most important question. The targeting efficiency of wo step targeting delivery strategy was limited by the pretargeting and the second targeting abilities, and the recognition and the targeting efficiency could not reach 100 %. I think one step targeting strategy with anti-HER2 modification could achieve an improved targeting efficiency.

Reply 2

The issue was thoroughly addressed in the manuscript and the direct comparison of one-step strategy (with PLGA-DARP9_29) and two-step strategy (with PLGA-Bn and Bs-DARP9_29) was performed:

Half-maximal inhibitory concentration (IC50) of doxorubicin calculated for one-step DDS and two-step DDS was found to be:

  1. i) One step DDS: IC50 = 4972 ± 1965 nM;
  2. ii) Two-step DDS: IC50 = 43 ± 3 nM,

thus proving the more effective delivery of doxorubicin inside nanoparticles with two-step DDS in comparison with one-step DDS.

Comment 3

Two step targeting delivery strategy faced a limited in vivo application. The authors were suggested to give a detail discussion.

Reply 3

The potential of two-step DDS based on barnase*barstar interaction for in vivo use is thoroughly discussed as follows:

“Considering potential in vivo applications of barnase*barstar protein pair for diagnostic and therapeutic applications, it should be emphasized that this system significantly outperforms other protein complementary systems in different aspects. Namely, it was previously shown that this protein system possesses unique stability under severe conditions (low pH, high temperature, and presence of chaotropic agents) in contrast to other known complementary protein systems (streptavidin*biotin, antibody*antigen, and protein A*immunoglobulin) [34]. This unique stability makes us believe that this complex will not be degraded in the bloodstream, where much milder conditions are realized, and will allow the self-assembly of different structures in vivo. Which is more important, these proteins are not presented in mammals, thus making them excellent candidates for use in the bloodstream without any interaction with en-dogenic components of blood in contrast to, e.g. streptavidin*biotin pair – biotin is a B7 vitamin that is presented in blood in the relatively high amount [33,65]. Moreover, these proteins possess unique proteolytic stability and are not subject to rapid degradation by proteases in the bloodstream. In contrast, various systems based on DNA/RNA self-assembly undergo rapid degradation due to nucleases presented in serum.

These features made possible the successful use of barnase*barstar protein pair for the in vivo delivery of radiolabeled anti-HER2 trimer complexes, namely 4D5 scFv–barnase*barstar, in Balb/c Nu/Nu mice reaching more than 8% of anti-HER2 complex accumulation in HER2-positive tumors [32]. Moreover, it was shown that the barnase*barstar pair is an effective tool for switchable targeting of solid HER2-positive tumors by CAR-T cells in vivo [48], thus reasonably confirming the efficacy and versatility of the barnase*barstar interface for the wide range of biological applications in vitro and in vivo that require the self-assembly of different structures in different conditions”.

Comment 4

The writing should be thoroughly improved, and the manuscript was really hard to understand.

Reply 4

The grammatical mistakes and typos were corrected.

Comment 5

The anti-proliferation abilities of the pretargeting moieties should be evaluated.

Reply 5

The anti-proliferation abilities of the pre-targeting moieties was studied and presented in Fig. S2 as follows:

“We demonstrated the efficiency of two-step DDS for the targeted delivery to cancer cells. For cancer cell pre-targeting, DARPin9_29 fused with barstar was selected since this protein, in contrast to, DARPin9_29 fused with barnase (Fig. S2) does not exhibit significant cytotoxicity and can be further used in vivo as pre-targeting mole-cule in large doses without any significant side effects. In contrast to DARPin9_29 fused with barstar, DARPin9_29 fused with barnase exhibit its own anti-cancer effect [48] and is not the best option for the pre-targeting strategy implying the use of first non-toxic component injected in a large dose before the injection of the second toxic component.

Figure S2. Cytotoxicity of DARP9_29-barnase and DARP9_29-barstar. BT-474 cells were incubated with proteins at different concentration and after 7 days of cultivation the MTT-test reflecting the number of viable cells was performed”.

Round 2

Reviewer 2 Report

The manuscript has been improved according to the comments, but some specific issues should be addressed to highlight the importance of this work.

1. Two-step targeting strategy exhibits an improved delivery efficiency in this work. However, one step targeting of HER2 may be more effective in consideration of the recognization and targeting efficiency of HER2 and DARPin9_29. And a clear discussion is suggested to be provided for the mechanism of the improved efficiency.

2. The IC50 of free doxorubicin is 441 nM, and the IC50 of two step DDS is 43 nM. As we know, doxorubicin can penetrate into cells and accumulate in necleus through diffusion process with high efficiency. Moreover, the nanoparticled PLGA will impede the intracellular delivery of nanomedicine by endocytosis process, and doxrubicin should be released from PLGA to exert antiproliferation effects. In most cases, doxrubicin exhibits lower IC50 than that of targeted nanomedicine. A clear discussion should be provided, and the cellular uptake behaviors of doxrubicin and two step DDS should be provided and quantitatively analyzed.

Author Response

Comment 1

Two-step targeting strategy exhibits an improved delivery efficiency in this work. However, one step targeting of HER2 may be more effective in consideration of the recognization and targeting efficiency of HER2 and DARPin9_29. And a clear discussion is suggested to be provided for the mechanism of the improved efficiency.

Reply 1

We thank the reviewer for this valuable comment and thoroughly discussed this issue within the Discussion section as follows:

“The development of targeted nanoparticle-based DDS is one of the most rapidly developing directions of modern biomedicine. Nanoparticles exhibit unique properties as therapeutic compounds and often outperform their molecular counterparts. The use of full-size IgG for targeted delivery to cells is already getting outdated today in favor of artificial scaffold polypeptides. Indeed, the small size, high stability, and ease of biotechnological production allow scaffolds to be effectively used as tools for targeted delivery [16,24]. In particular, the use of scaffolds (for example, DARPins, 14–16 kDa) makes it possible to equip the surface of nanoparticle by targeting molecules with high density in contrast to, e.g., big IgG (150 kDa) to get an optimal affinity in the nanoparticle*cell interaction. On the other hand, the small size of DARPins conjugated to the surface of therapeutic nanoparticles, can lead to steric hindrance during target recognition.

In particular, we previously performed a direct comparison of HER2-overexpressing cell-targeting abilities of nanoparticle*DARPin9_29 and nanoparticle*anti-HER2 IgG for two nanoparticle types: SiO2 fluorescent nanoparticles and SiO2 magnetic nanoparticles [41]. We showed that direct conjugation of DARPin9_29 to nanoparticle surface is absolutely ineffective for HER2 targeting in contrast to nanoparticle*anti-HER2 IgG. However, cell targeting mediated with proteinaceous barnase*barstar interface showed a significantly higher efficiency compared to both nanoparticle*DARPin9_29 and nanoparticle*anti-HER2 IgG. The nanoparticle cell uptake was more than 3 times higher for the two-step approach in comparison with the one-step one [41]. Thus, the barnase*barstar interface significantly enhanced the particle binding efficiency and increased the cell uptake under equal conditions”.

Comment 2

The IC50 of free doxorubicin is 441 nM, and the IC50 of two step DDS is 43 nM. As we know, doxorubicin can penetrate into cells and accumulate in necleus through diffusion process with high efficiency. Moreover, the nanoparticled PLGA will impede the intracellular delivery of nanomedicine by endocytosis process, and doxrubicin should be released from PLGA to exert antiproliferation effects. In most cases, doxrubicin exhibits lower IC50 than that of targeted nanomedicine. A clear discussion should be provided, and the cellular uptake behaviors of doxrubicin and two step DDS should be provided and quantitatively analyzed.

Reply 2

We thank the reviewer for this valuable comment and thoroughly discussed this issue within the main text as follows:

“ The quantitative analysis of obtained results demonstrates that two-step DDS is much more effective than one-step DDS in terms of IC50. Namely, cells exposed to non-targeted PLGA nanoparticles or one-step DDS were not affected by the cytotoxic properties of doxorubicin loaded inside nanoparticles, and cells survived by more than 82 % even at the highest concentrations of PLGA, namely, at 1 g/L (Fig. 10b). However, the delivery of doxorubicin via two-step DDS decreased its IC50 by 10.3 times vs. free doxorubicin and more than 100 times vs. one-step DDS.

Usually, the incorporation of doxorubicin into nanoparticle structure does not significantly decrease its IC50 or even increase it [52,53]. The main goal of such incorporation is to minimize the cardiac toxicity of doxorubicin and increase its tumor accumulation [54–56]. However, for targeted nanoparticles, this trend is not always observed. It was previously shown that the decoration of PLGA nanoparticle surface with targeting peptides can decrease the IC50 of doxorubicin by more than 1 order of magnitude. Namely, the IC50 was shown to be equal to 5745 ± 2651 nM, 92 ± 55 nM, and 331 ± 37 nM for non-targeted doxorubicin-loaded PLGA, anti-EGFR PLGA, and free doxorubicin, respectively [57]. Another example showed that anti-HER2 chitosan nanoparticles loaded with doxorubicin demonstrate IC50 equal to 377 nM in contrast to 1235 nM for free doxorubicin [58]. Such unexpected results may be explained by the alterations in mechanisms of doxorubicin action in cells when it is delivered with a nanocarrier.

The primary doxorubicin toxicity mechanism is based on the intercalation with DNA base pairs in the nucleus, thus blocking topoisomerase II and causing DNA breakage, leading to the inhibition of DNA, RNA, and protein biosynthesis processes [45]. However, when doxorubicin is delivered within the targeted nanoparticle to cell-membrane receptors, other mechanisms of its cytotoxicity may become more pronounced. Namely, the oxidation of doxorubicin to semiquinone with the release of reactive oxygen species leading to lipid peroxidation and membrane damage is most likely to have a more pronounced cytotoxic effect when it occurs near the cell membrane with the slow release of doxorubicin from nanoparticles in contrast to free doxorubicin freely dispersed in the cytoplasm and nucleus as a result of diffusion processes”.